# Impaired Ghrelin Signaling Does Not Lead to Alterations of Anxiety-like Behaviors in Adult Mice Chronically Exposed to THC during Adolescence

**DOI:** 10.3390/biomedicines11010144

**Published:** 2023-01-06

**Authors:** Matija Sestan-Pesa, Marya Shanabrough, Tamas L. Horvath, Maria Consolata Miletta

**Affiliations:** 1Department of Comparative Medicine, Yale University School of Medicine, New Haven, CT 06520, USA; 2Program in Integrative Cell Signaling and Neurobiology of Metabolism, Yale University School of Medicine, New Haven, CT 06520, USA; 3Larsson-Rosenquist Foundation Center for Neurodevelopment, Growth and Nutrition of the Newborn, Department of Neonatology, University of Zurich and University Hospital Zurich, 8006 Zurich, Switzerland

**Keywords:** ghrelin, tetrahydrocannabinol (THC), GHSR signaling, late adolescence, endocannabinoid system

## Abstract

As marijuana use during adolescence has been increasing, the need to understand the effects of its long-term use becomes crucial. Previous research suggested that marijuana consumption during adolescence increases the risk of developing mental illnesses, such as schizophrenia, depression, and anxiety. Ghrelin is a peptide produced primarily in the gut and is important for feeding behavior. Recent studies have shown that ghrelin and its receptor, the growth hormone secretagogue receptor (GHSR), play important roles in mediating stress, as well as anxiety and depression-like behaviors in animal models. Here, we investigated the effects of chronic tetrahydrocannabinol (THC) administration during late adolescence (P42–55) in GHSR (GHSR ^−/−^) knockout mice and their wild-type littermates in relation to anxiety-like behaviors. We determined that continuous THC exposure during late adolescence did not lead to any significant alterations in the anxiety-like behaviors of adult mice, regardless of genotype, following a prolonged period of no exposure (1 month). These data indicate that in the presence of intact or impaired ghrelin/GHSR signaling, THC exposure during late adolescence has limited if any long-term impact on anxiety-like behaviors in mice.

## 1. Introduction

Adolescence is the developmental period of transition between childhood and adulthood, on average starting at age 12 and ending at age 18 [1,2]. This period is marked by significant neuroplasticity in the prefrontal cortex and limbic regions, two brain regions involved in development of adult behavior and cognitive functions [1,3]. 

Cannabis use among adolescents is very high, with 9.4% of 8th graders, 23.9% of 10th graders, and 36.5% of 12th graders reporting cannabis use in the last 12 months in 2016 [4]. This event is concerning as cannabis abuse can lead to persistent cognitive impairments in learning, attention and memory [5,6,7,8,9,10]. Moreover, early cannabis use before 16 years of age increases the risk of developing psychiatric disorders, including anxiety-related symptoms [11,12,13]. Anxiety appears to be the most common complication arising from heavy cannabis use, with up to 20% of cannabis users experiencing anxiety [14] while the prevalence of anxiety in the general population is estimated to be around 6–17% [15]. 

The primary psychoactive component of cannabis is delta-9-tetrahydrocannabinol (THC). The biological effects of THC are mainly mediated by members of the G protein-coupled receptor (GPCR) family, such as cannabinoid receptors (CB1R and CB2R).

The cannabinoid receptors together with their naturally occurring ligands (anandamide and 2-arachidonoyl glycerol) and the enzymes responsible for their biosynthesis constitute the endocannabinoid system [16,17]. This system plays a critical role in the maturation of brain circuits during adolescence by regulating excitatory and inhibitory neurotransmission [18]. Further, CB1R expression increases dramatically in regions such as the prefrontal cortex, striatum, and hippocampus [19]. Imaging studies have shown decreased cortical thickness in the right superior prefrontal cortex (PFC), bilateral insula and bilateral superior cortices in adolescent cannabis users compared to adolescents who do not use cannabis [20], as well as a decrease in volume of the right medial orbitofrontal cortex [21] and bilateral hippocampus [22,23].

Ghrelin is a hormone mainly produced in the gut [24]. It stimulates potent orexigenic effects through metabolic homeostatic regulatory mechanisms in the hypothalamus and by increasing food reward and motivation through mesolimbic activation [25,26]. Ghrelin mediates both peripheral and central physiological functions through the growth hormone secretagogue receptor (GHSR) [27]. Ghrelin’s role in regulating mood is very complex and it has a dual role in regulating anxiety. In some cases, injecting ghrelin centrally increased anxiety-like behaviors assessed by elevated plus maze [28], while other reports suggest the opposite effect, with ghrelin injections showing a decrease in anxiety-like behaviors as assessed by elevated plus maze [29]. This discrepancy might be related to the timing of the behavioral experiments. Another factor that contributes to modulateing ghrelin’s effect on behavior is food availability, with ghrelin increasing locomotion in the absence of food [30] and decreasing locomotion in the presence of food [31]. Findings in ghrelin knockout mice also demonstrate the controversial relationship between ghrelin and anxiety. Ghrelin knockout (Ghr ^−/−^) mice appear to be less anxious than their wild-type counterparts under non-stressed conditions, but display more anxious behavior under mild stress conditions (15 min restraint) [32]. Of note is that stress increases ghrelin and corticosterone concurrently. GHSR and ghrelin knockout mice showed decreased plasma levels of corticosterone after chronic social defeat stress and acute restraint stress, as well as increased anxiety-like behavior [32,33]. Taken together, these findings suggest that ghrelin and GHSR are important for the ability of animals to cope with anxiety-inducing stressors. GHSR and the cannabinoid CB1R are expressed within overlapping brain regions that are crucial for feeding (hypothalamus), reward and motivation (Ventral tegmental area/VTA, nucleus accubens/NAC). Both systems mutually interact to a significant extent in the regulation of homeostatic as well as hedonic food intake [34,35,36,37]. Further, systemic pretreatment with the CB1R antagonist rimonabant significantly reduced intracerebroventricular ghrelin-induced NAC dopamine release and hyperlocomotion in mice [38]. Despite this knowledge, there are limited data on the mutual role of cannabis and ghrelin in promoting anxiety-like behaviors. Therefore, we aim to test the way in which GHSR ^−/−^ mice (and their wild-type counterparts) would respond to chronic THC administration during adolescence. To investigate the long-term effects of THC on behavior relating to anxiety, we exposed the animals to 10 mg of THC daily (via pulmonary route) during sexual maturation (6–8 weeks old mice), which roughly corresponds to adolescence in humans. After 14 days of THC administration, animals (male and female mice) could recover for additional 4 weeks. At 12 weeks of age, behavioral testing was performed to evaluate any long-term effects from THC administration (Figure 1A). 

## 2. Materials and Methods

### 2.1. Materials

To closely mimic human THC consumption, we used a formulation of THC (3.62% THC, 6.47% tetrahydrocannabinolic acid (THCA), a total of 101 mg/ml of THC) with a minimal content of terpenes (β-myrcene 0.06%, β-caryophyllene 0.64%, humulene 0.39%; a total of 1.09% terpenes) dissolved in Polyethylenglycol (PEG 400), designed for use with a commercially available vaporization apparatus. 

PEG 400 with terpenes was used as the vehicle for the control group. Connecticut Pharmaceutical Solutions, LLC, Portland, CT, USA (a state-licensed grower) provided the compounds through the Connecticut Medical Marijuana Research Program.

### 2.2. Animals

The Institutional Animal Care and Use Committee of Yale University approved all experiments (protocol code 2019-07942). Mice were kept under standard laboratory conditions with free access to standard chow food and water except during behavioral testing. Mice were generated by breeding C57BL/6J (n. 000664 Jackson Lab) with GHSR ^−/+^ mice in order to obtain an F1 generation of heterozygous GHSR knock-out animals. These progenies were subsequently used to generate GHSR ^+/+^ (WT) and GHSR ^−/−^ (KO) animals used in this study. All animals were generated, bred and weaned by our laboratory and housed in the same animal room. Further details can be obtained from our previous publication [39]. THC was administered to animals from 6 to 8 weeks of age and behavioral testing was performed at 12 to 13 weeks of age (Figure 1A). Animals were placed into 2 treatment groups (vehicle and 10 mg THC) for each genotype (wild type and knock-out). Since we did not have a strict hypothesis of what sort of difference we can expect, we used the “resource equation” method for defining our sample size, a commonly used way of establishing sample size in exploratory studies [40]. We did not control for the estrous cycle in female mice.

### 2.3. THC Administration 

Most adolescents smoke cannabis; therefore, we decided to mimic smoking as a method of administration of THC. We used commercially available vaporization equipment for marijuana administration, commonly used by marijuana consumers. Previously described experiments used the Volcano^®^ Vapourization device (Storz and Bickel, GmbH and Co., Tuttlingen, Germany) to administer ethanol-dissolved THC to lab animals in a consistent and reproducible manner, and presented similar dose-dependent and time-dependent changes for both pulmonary and parenteral administration [41,42]. We followed their procedure, except for the following modifications. In our study, the THC containing formulation and vehicle were vaporized at 175 °C in order to avoid excessive formation and vaporization of cannabinol (CBN, a psychoactive metabolite of THC which may confound results) and reaching the flashpoint of the vehicle (PEG400, 250 °C). 

THC and vehicle were administered under a chemical hood to prevent cross-contamination due to leakage of vapor, as well as maintaining a consistent experimental environment (Figure 1B). Mice were placed, in groups of 2–4, inside a closed chamber (33 cm × 20.3 cm × 10.2 cm) with valves and tubing on two of the narrower sides. To further minimize vapor escaping, on one side the tubing led to an improvised activated charcoal trap, leading to an activated charcoal filter, leading to the vacuum line (Figure 1C,D). On the other side, tubing was open-ended with a Volcano Vaporizer mouthpiece fixed to it (Figure 1E). The mouthpiece was used to release the seal on the balloons, which were filled with a vapor containing THC (or vehicle; note that the content of the balloons is mostly air so that the animal was always normoxic while in the chamber). Parafilm was used to seal the connection making it airtight (Figure 1F). Animals were exposed to the vapor for 5 min, with half of the balloon being emptied at the beginning and the other half being emptied after 2 or 3 minutes of exposure. We separated the evacuation of the vapor-filled balloon into 2 parts to prevent excessive leaking of vapors caused by increasing pressure inside of the box which exacerbates cracks in the boxes’ seal during balloon evacuation. The vacuum pump line was not efficient enough at maintaining a stable pressure to avoid side leakage and posed a danger to the animals inside the box if activated when they were still inside. The leakage was minimized to our satisfaction by splitting the balloon evacuation into 2 parts. After the exposure, animals were quickly removed from the chamber, and the vacuum line was turned on to remove any residual vapor. The inside of the box was cleaned with 70% ethanol between each group of animals.

### 2.4. Behavioral Assessments 

Open field and elevated zero maze were used to establish behavioral phenotypes induced by THC administration. Behavioral testing was performed during the light phase of the cycle from 1 pm to 7 pm.

#### 2.4.1. Open Field

The open field apparatus (Stoelting Company, Wood Dale, IL, USA) is a square, polyurethane box (35.5 cm × 35.5 cm × 30 cm). The animal was placed in the center of the apparatus. General locomotion parameters (distance traveled, locomotion speed, time mobile) and parameters relating to anxiety (freezing time; time spent, distance travelled, and entries into central and periphery zones) were recorded for 10 min. The apparatus was cleaned with 70% ethanol after each animal’s exposure. ANY-Maze software (Stoelting Company, Wood Dale, IL, USA) was used to record and analyze the behavioral data.

#### 2.4.2. Elevated Zero Maze

The elevated zero maze apparatus is an elevated (60 cm high) ring-shaped runway (5 cm wide), with 2 equally sized (25% of the runway length) sections closed off by walls (40 cm high) opposite each other. The other two sections are open (Stoelting Company, Wood Dale, IL, USA). The maze was equally illuminated in all four sections. Mice were placed in the center of one of the open sections, facing one of the closed sections, and allowed to explore the maze for 5 min. The apparatus was cleaned with 70% ethanol after each animal’s exposure. ANY-Maze SoftwareTM (Stoelting Company, Wood Dale, IL, USA) was used to record and analyze the behavioral data.

### 2.5. Statistics

GraphPad Prism 8.0 and Microsoft Excel 14.4.2 were used to analyze data and plot figures. Since our goal was to compare the means of more than two groups of animals, while controlling for two variables (genotype and THC treatment), two-way ANOVA was used to analyze the results. When results were significant, a multiple comparison test was performed, comparing the means of all groups, regardless of the variable in question. Data are expressed as the mean ± standard error of the mean (SEM), and a *p* value ≤ 0.5. was considered statistically significant.

## 3. Results

### 3.1. Open Field

The open field exploration test represents a unique opportunity to systematically assess novel environment exploration and general locomotor activity and provides initial screening for anxiety-related behaviors in rodents [43]. Two factors influence anxiety-like behaviors in the open field. The first is social isolation resulting from the physical separation from cage mates when performing the test. The second is the stress created by the brightly lit, unprotected, novel test environment [44,45]. To assess the behavioral effects of adolescent exposure to THC, we treated female and male mice of two different genotypes (GHSR ^+/+^ WT) and GHSR ^−/−^ (KO) during adolescence. We assessed anxiety-like behaviors in young adulthood (Figure 1A).

In our experimental setting, parameters that evaluate general locomotion showed no differences amongst the groups, suggesting locomotion was unaffected by THC exposure (Figure 2A–E). Further, we did not observe significant differences in the in the open field parameters indicative of anxiety-like behaviors in all groups independently of genotype and/or treatment (Figure 3A–E). For all main effects and interaction, *p* ≥ 0.05.

### 3.2. Elevated Zero Maze

Elevated zero maze is the master test for assessing anxiety-like behaviors in mice. The test exploits the natural tendencies of mice to explore novel environments [46].

We did not observe significant differences in the willingness of mice to explore open environments in all groups independently of genotype and/or treatment. For all main effects and interaction, *p* ≥ 0.05 Figure 4A–E.

## 4. Discussion

In this study, we demonstrated that administration of THC to adolescent mice does not cause anxiety-like behavior in adult mice nor affect basic locomotor activity. Further, we showed that impairment of the ghrelin signaling through the knockout of the GHRS does not confer an increased risk of developing THC induced anxiety in adult mice. Our results are consistent with previous reports in rodent models that concluded that prolonged adolescent THC exposure in mice does not have substantive negative impacts on several mPFC-mediated behaviors [47,48,49,50]. In particular, Chen et al. [49] treated 28-day-old C57BL6/J mice of both sexes for three weeks with 3 mg/kg THC (daily intraperitoneal injections i.p.). One week after recovery, they analyzed several cognitive behaviors and detected little effect on anxiety-like behaviors. In another study, Zuo et al. [48] treated female and male mice with 10 mg/kg of THC in early adolescence (1 i.p. for 21 consecutive days during postnatal weeks 5–7) and assessed the impact on anxiety-like behaviors two weeks later. Their behavioral analysis demonstrated that adolescent exposure to THC in mice led to long-term impairments in object recognition, memory and social interaction, but not in anxiety-like behaviors. The experimental evidence on long-term effect of cannabis exposure during adolescence includes cannabidiol (CBD) as well, a non-intoxicating phytocannabinoid. Prolonged adolescent CBD exposure had no detrimental effects on locomotor activity in the open field and anxiety-like behaviors on the elevated plus maze in male and female C57BL/6J mice treated for 20 days mg/kg with two daily i.p. injections of CBD (20 mg/kg) [47].

Our exposure period starting at postnatal day (PND) 42 and ending at PND 55 represents the mouse brain development period similar to human adolescence [51]. Earlier findings identified this period as the critical time window for persistent detrimental effects of cannabis misuse [52,53]. Cannabis mainly acts on the developing cerebral cortex, especially the medial prefrontal cortex, a late-developing brain region whose volume decreases dramatically during adolescence as it undergoes synaptic refinement [54]. 

Interestingly, our results show that impairing ghrelin signaling through GHSR knockout does not affect the long-term outcome of the THC treatment. Since ghrelin and THC often act synergistically in many pathways [37], the results on GHRS KO mice further corroborate the lack of significant long-term alterations of anxiety-like behavior induced by THC in our experimental setting.

Preclinical studies collectively suggest that ghrelin/GHSR reinforces the action of cannabinoids and CB1 agonists [55,56]. 

These studies imply the potential interaction of the ghrelin signaling with other neurotransmitter systems (the endocannabinoid, and GABA systems) within the NAC in the reinforcing effects of cannabinoids [57]. 

For example, the GHSR antagonist JMV2959 significantly reduced several parameters of cannabinoid reward and attenuated cannabinoid intake and drug-seeking behavior [56]. 

The ghrelin receptor can interact with the CB2 cannabinoid receptor in both heterologous cells and cells of the central nervous system [58].

Overall, the long-term outcomes of cannabis exposure during adolescence are complex and can result from multiple factors. Early life interferences such as maternal deprivation or immune system activation could increase the vulnerability to cannabis-related developmental insults [59,60]. Investigations into the neurodevelopmental exposure to THC in translational animal models could provide insights into various neural pathways and biomarkers involved in THC-related pathological outcomes, identifying potential molecular targets for novel pharmacotherapeutic approaches [61].

### Limitations and Future Studies

In this study, we used the vaporization method to administer THC. Compared to i.p. injections, the pharmacodynamics of the THC following vaporization is less known. Future research should establish the dose and plasma level relationship for the vaporization method using the liquid chromatography–mass spectrometry (LC/MS) method, as performed before [41,42]. Moreover, we used a lower temperature for the vaporizations, thus we should assume that less of the material was vaporized compared to the original papers [41,42]. Most of our animals exposed to THC displayed instant changes in behavior, such as headshakes (similar to hallucinogen-induced headshakes), hyperactivity, or mild somnolence. Considering that the experiments on rats, from the original study, showed a hyperactivity phenotype at 1 mg and hypomobility/somnolence with 10 mg of vaporized THC, we assumed that in our experiments, active THC reached levels between these two reported doses.

Changing conditions such as dose, the time of THC exposure, concomitant stress exposure, and presence/lack of food could clarify if there are any relevant conditions under which THC can significantly alter long-term anxiety-like behaviors with or without an intact Ghrelin/GHRS signaling. Male and female mice might be affected differently by THC exposure since the literature suggests that females may be more vulnerable to THC’s effect on anxiety [62,63]. To address the possible sex differences and ghrelin’s role in them, THC exposure should be coupled with a variety of adjunct treatments, such as sex hormone inhibitors and ghrelin. Lastly, additional behavioral tests, such as pre-pulse inhibition, marble burying, and tail suspension, should be employed to investigate whether THC exposure in late adolescence affects behaviors related to sensory gating, compulsiveness, and mood regulation.

## Figures and Tables

**Figure 1 biomedicines-11-00144-f001:**
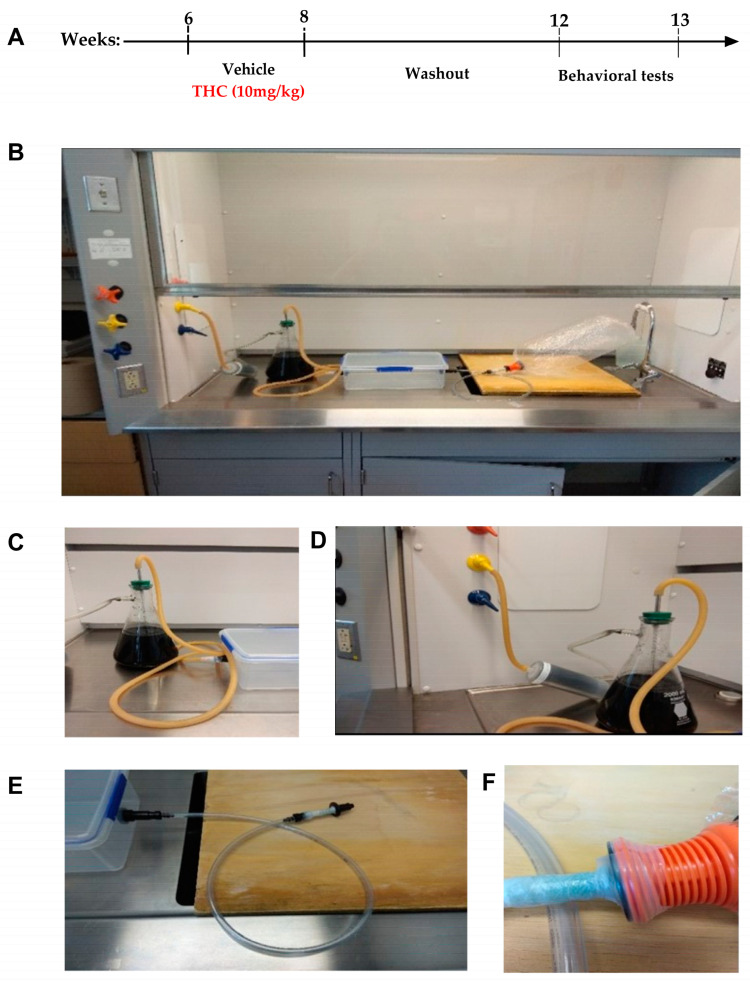
Experiment design and equipment for THC administration. (**A**) Experimental design, time course for THC (or vehicle) administration and behavioral testing. (**B**) Entire apparatus used to administer THC and vehicle under a chemical hood. (**C**) Tubing leading from administration box to activated charcoal trap. (**D**) Tubing leading from activated charcoal trap to the activated charcoal filter, which then leads to the vacuum line. (**E**) Open-ended tube with Volcano mouthpiece attached. (**F**) Balloon attached to the open-ended mouthpiece, sealed with parafilm.

**Figure 2 biomedicines-11-00144-f002:**
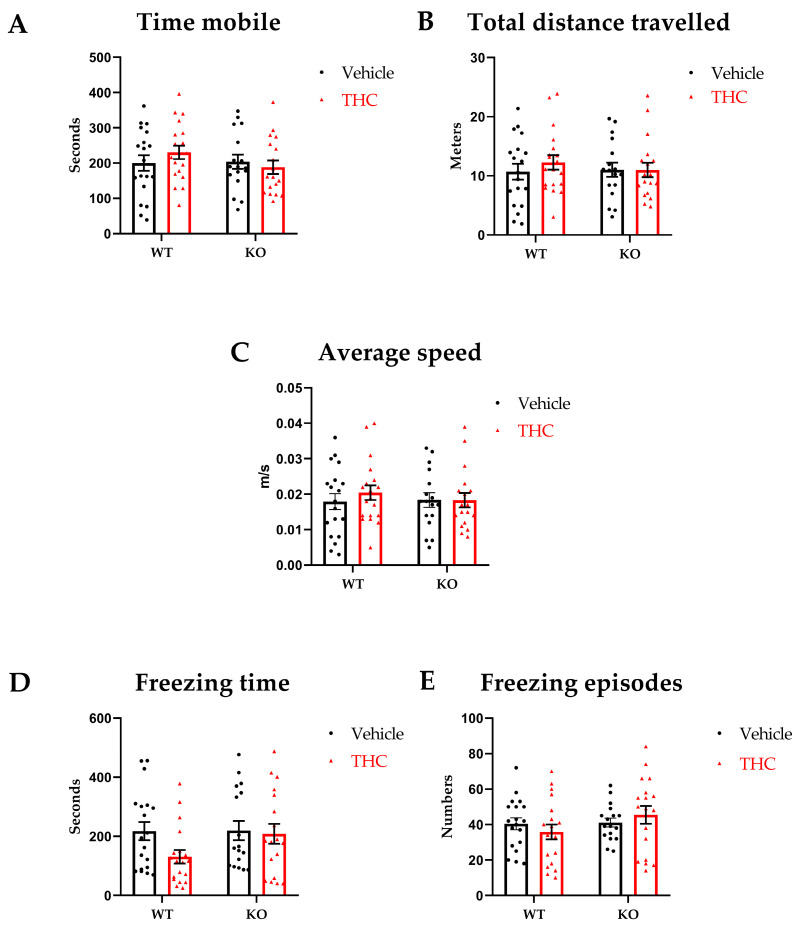
Open field, general locomotion. WT-vehicle *n* = 19, WT-THC *n* = 18, KO-vehicle *n* = 19, KO-THC *n* = 18; Male and female mice were included. (**A**) Time mobile. (**B**) Total distance travelled. (**C**) Average speed. (**D**) Freezing time. (**E**) Number of freezing episodes. Data are expressed as mean ± SEM Two-way ANOVA plus post hoc comparison test.

**Figure 3 biomedicines-11-00144-f003:**
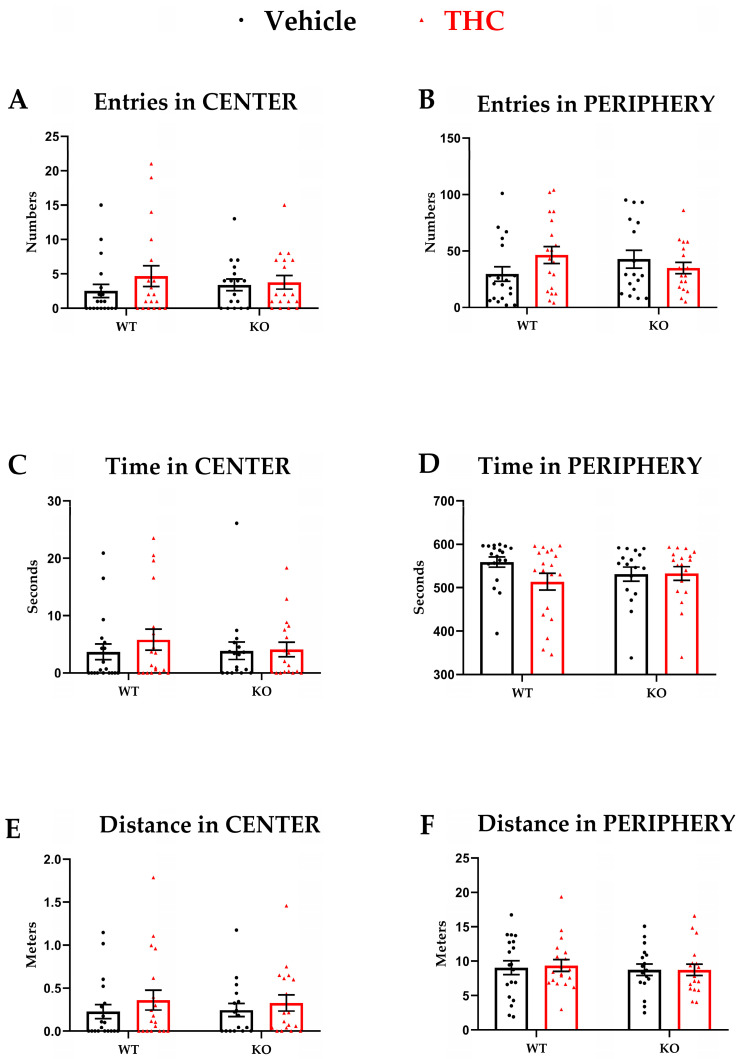
Open field, anxiety-related parameters. WT-vehicle *n* = 19, WT-THC *n* = 18, KO-vehicle *n* = 19, KO-THC *n* = 18; Male and female mice were included. (**A**,**B**) Number of entries in the central or peripheral zone. (**C**,**D**) Time spent in central or peripheral zone. (**E**,**F**) Total distance traveled in central or peripheral zone. Data are expressed as mean ± SEM. Two-way ANOVA plus post hoc comparison tests.

**Figure 4 biomedicines-11-00144-f004:**
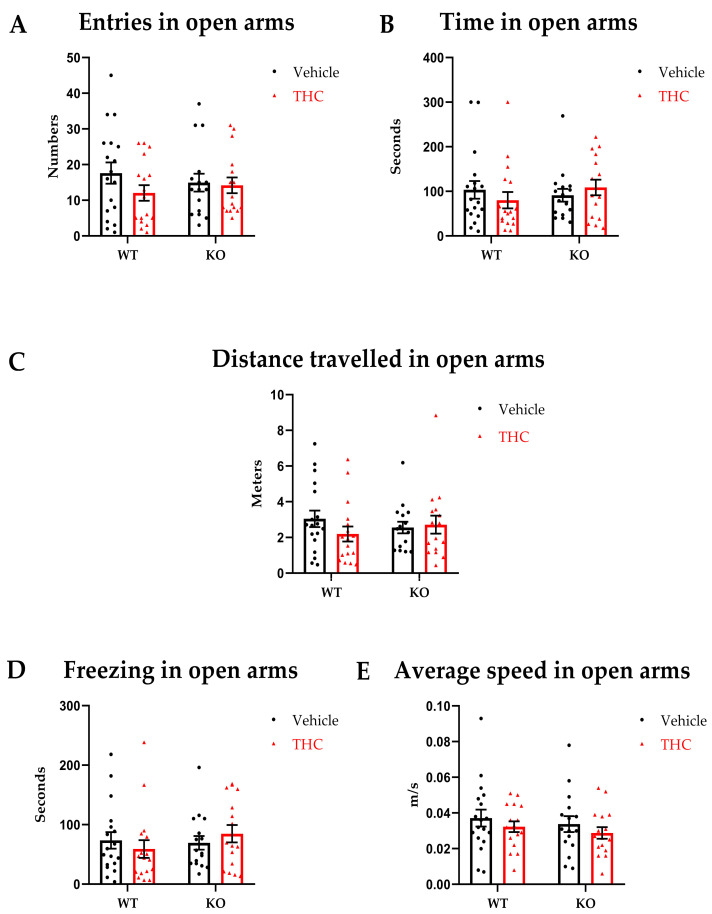
Elevated Zero Maze. WT-vehicle *n* = 19, WT-THC *n* = 18, KO-vehicle *n* = 19, KO-THC *n* = 18. Male and female mice were included. (**A**) Entries in open arms. (**B**) Time spent in open arms. (**C**) Distance travelled in open arms. (**D**) Average speed in open arms. (**E**) Time spent freezing in open arms. Data are expressed as mean ± SEM. Two-way ANOVA plus post hoc comparison test.

## Data Availability

The data that support the findings of this study are available from the corresponding author upon reasonable request.

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
