# Peer review of "Impaired Ghrelin Signaling Does Not Lead to Alterations of Anxiety-like Behaviors in Adult Mice Chronically Exposed to THC during Adolescence"

_biomedicines, 2023, doi:10.3390/biomedicines11010144_

Round 1
Reviewer 1 Report
In this manuscript Sestan-Pesa et al. present the methodology and the results of some experimental researches evaluating the chronic administration of tetrahydrocannabinol during late adolescence, in GHSR-/- knockout mice and their wild type littermates in relation to anxiety-like behaviors in open field and elevated zero maze models.
The present style of manuscript cannot be accepted for publication due to several reasons described below. I recommend the authors the major revision to blush the manuscript up.
After reading the manuscript, the following doubts and suggestions have arisen.
- the topic is not a novelty in the field.
- the introduction section should be more complete, providing supplementary background in the field.
- the results obtained should be compared with those achieved by other researchers and discussions should be significantly detailed (see:
· Zuo, Y et al. Chronic adolescent exposure to cannabis in mice leads to sex-biased changes in gene expression networks across brain regions. Neuropsychopharmacol. 2022; 47, 2071–2080
· Kaplan JS et al., Cannabidiol Exposure During the Mouse Adolescent Period Is Without Harmful Behavioral Effects on Locomotor Activity, Anxiety, and Spatial Memory. Front Behav Neurosci. 2021 Aug 26;15:711639.
· De Felice M et al. Reversing the Psychiatric Effects of Neurodevelopmental Cannabinoid Exposure: Exploring Pharmacotherapeutic Interventions for Symptom Improvement. Int. J. Mol. Sci. 2021, 22(15), 7861;
· Portugalov A et al. Do Adolescent Exposure to Cannabinoids and Early Adverse Experience Interact to Increase the Risk of Psychiatric Disorders: Evidence from Rodent Models. Int J Mol Sci. 2021;22(2):730.
· Chen HT et al. Adolescent Δ9-Tetrahydrocannabinol Exposure Selectively Impairs Working Memory but Not Several Other mPFC-Mediated Behaviors. Front Psychiatry. 2020;11:576214.
- the authors need to develop argumentation in depth based on the current understanding and the findings of the results obtained, presenting the potential, the weakness and limitation, and future research direction, among others. Authors should try to explain the theoretical implication as well as the translational application of their research.
- the abbreviations should be mentioned in the text (PFC, THCA, PEG, SEM);
- the abbreviations should be expanded in the first appearance. The explanation of the abbreviation should be used only once in the text and should not be repeated, in order to decongest the text and facilitate the understanding of the information transmitted.
- missing information (city, country) about the companies producing some devices used in the research (open field apparatus, elevated zero maze, HPLC, UV detector);
- the authors should upgrade the references;
- different fonts were used in the text and in figures;
- the quality of figures 2,3,4 should be improved;
- spelling check of the text is mandatory (line 173 - elevate instead of elevated; line 272 – LQ/MS instead of LC/MS, and many others);
- English including grammar, style and syntax, should be improved through the professional help from English Editing Company for Scientific Writings.
Reviewer 2 Report
The submitted manuscript analyses the behavior of adolescent ghrelin ko mice vs controls following pulmonary exposure to THC. The manuscript is quite preliminary, lacks any molecular, histological or electrophysiological data and does not register any THC dependent effects. Furthermore, in discussion section the authors underline their negative results may be the consequence of poor sample size. Taken together, the manuscript is interesting, but too much preliminary for Biomedicines.
In addition:
Specify the sex of the animals and the number of animals/sex. It is not clear why preliminary experiments were carried out in females and experiments in males (or both?). Please, specify.
Reference list is not formatted accordingly the journal style.
Round 2
Reviewer 1 Report
The authors have significantly revised the manuscript addressing the concern raised. I consider it could be accepted for publication in this journal.
Reviewer 2 Report
The authors have deeply improved their manuscript; they re-wrote several part of the manuscript, included more results and better presented and discussed their data.
In the current version I endorse the manuiscript for publication